# DYNAMIC CLASSIFIER-FREE DIFFUSION GUIDANCE VIA ONLINE FEEDBACK

**Pinelopi Papalampidi**[1][*], **Olivia Wiles**[1][*], **Ira Ktena**[2][*][†], **Aleksandar Shtedritski**[3][†],
**Emanuele Bugliarello**[1], **Ivana Kajić**[1], **Isabela Albuquerque**[1], **Aida Nematzadeh**[1]
[1]Google DeepMind, [2]Ellison Institute of Technology, [3]University of Oxford
pinelopi@google.com

## ABSTRACT

Classifier-free guidance (CFG) is a cornerstone of text-to-image diffusion models, yet its effectiveness is limited by the use of static guidance scales. This "one-size-fits-all" approach fails to adapt to the diverse requirements of different prompts; moreover, prior solutions like gradient-based correction or fixed heuristic schedules introduce additional complexities and fail to generalize. In this work, we challenge this static paradigm by introducing a framework for dynamic CFG scheduling. Our method leverages online feedback from a suite of general-purpose and specialized small-scale latent-space evaluators—such as CLIP for alignment, a discriminator for fidelity and a human preference reward model—to assess generation quality at each step of the reverse diffusion process. Based on this feedback, we perform a greedy search to select the optimal CFG scale for each timestep, creating a unique guidance schedule tailored to every prompt and sample. We demonstrate the effectiveness of our approach on both small-scale models and the state-of-the-art Imagen 3, showing significant improvements in text alignment, visual quality, text rendering and numerical reasoning. Notably, when compared against the default Imagen 3 baseline, our method achieves up to 53.8% human preference win-rate for overall preference, a figure that increases up to to 55.5% on prompts targeting specific capabilities like text rendering. Our work establishes that the optimal guidance schedule is inherently dynamic and prompt-dependent, and provides an efficient and generalizable framework to achieve it.

## 1 INTRODUCTION

The remarkable progress in text-to-image synthesis, powered by diffusion models (Ho et al., 2020; Song et al., 2023), has unlocked unprecedented creative potential. However, generating images from diffusion models requires hundreds of sampling steps to achieve sufficient generation quality. Consequently, a critical frontier of research is not only in training more powerful models, but also in enhancing inference in terms of efficiency and controllability without the need for costly retraining.

A cornerstone of controlling the generation process at inference time is classifier-free guidance (CFG; Ho & Salimans 2022) which has become the de facto standard in image generation. CFG provides a mechanism to amplify the influence of the text prompt, allowing to trade diversity for stronger adherence to the conditioning signal via a single guidance scale. However, the guidance scale is typically either set to a single, static value for the entire generation process or is defined as a schedule depending only on the sampling timestep based on empirical observations (Kynkäänniemi et al., 2024; Chang et al., 2023; Sadat et al., 2023; Wang et al., 2024). In all cases, CFG is reduced to a "one-size-fits-all" strategy that overlooks the nuanced demands of different prompts during inference. For example, a prompt requiring complex compositional arrangements may need strong guidance for text alignment, whereas a prompt focused on a specific artistic aesthetic might benefit from lower guidance to preserve visual fidelity and diversity. We empirically validate this hypothesis and further find that generating specific, challenging attributes like legible text within an image often

---

[*]Equal contributions
[†]Work done while at Google DeepMind.

responds poorly to standard guidance strengths. This rigidity forces an undesirable compromise, where optimizing for one aspect (e.g., alignment) often degrades another (e.g., aesthetics).

In this paper, we challenge the notion of a static guidance scale in diffusion models. We hypothesize that the optimal trade-off between prompt alignment and visual quality is not fixed, but is a dynamic function of the prompt's content, the current generation stage, and the diffusion model itself. To realize this, we propose a framework that dynamically selects the optimal CFG scale using online feedback from efficient latent evaluators. We employ a suite of these evaluators to measure distinct generation capabilities: both general-purpose (alignment, visual quality) and specialized ones such as text rendering and numerical reasoning. Crucially, these evaluators operate directly on noisy latents within the diffusion process, providing rich feedback with negligible computational overhead.

We leverage a greedy search-based optimization at each sampling step to evaluate a discrete set of candidate CFG scales. We select the one that maximizes a composite score from our latent evaluators. This procedure generates a dynamic CFG schedule tailored specifically to each prompt and its evolving sample. Interestingly, the average trend of our schedules aligns with empirical heuristics from prior work (Kynkäänniemi et al., 2024; Wang et al., 2024), lending external validity to our approach. However, the key to our superior performance lies in the adaptability of our approach.

Our experiments on a text-to-image model similar to StableDiffusion (Rombach et al., 2022) across the Gecko (Wiles et al., 2024) and MS COCO (Lin et al., 2014) benchmarks demonstrate that our method improves both alignment and visual quality simultaneously. This stands in sharp contrast to prior methods, such as gradient guidance (Nichol et al., 2022; Kim et al., 2023) or fixed heuristic schedules (Kynkäänniemi et al., 2024; Sadat et al., 2023), which typically improve one aspect at the expense of the other.

To demonstrate the generality and scalability of our approach, we apply it to the SoTA Imagen 3 model (Team et al., 2024). On the challenging Gecko and GenAI-Bench (Li et al., 2024) prompt sets, human raters preferred generations from our method over the default Imagen 3 baseline in 53.6% and 53.8% of comparisons, respectively. The high quality of SOTA models also motivates extending our framework with more specialized, capability-based evaluators. By incorporating a human preference reward model, and text rendering and numerical reasoning specific evaluators, we achieve even more fine-grained control. For the MARIO-eval (Chen et al., 2023a) benchmark requiring legible text, and the GeckoNum (Kajić et al., 2024) one requiring counting skills, this specialized guidance boosts the human preference rate up to 55.5% and 54.1% over default sampling, respectively.

Our contributions can be summarized as follows:

- We propose a novel framework for dynamically optimizing the CFG schedule during generation and introduce a suite of latent evaluators that provide online feedback directly on noisy diffusion latents while increasing the computational requirements only by 1% in contrast to 400% for a pixel-space equivalent.
- We show that prior empirical observations on CFG schedules fail to generalize across different model families, prompt sets, and generation skills. In contrast, our method significantly improves sampling on both a StableDiffusion-equivalent model and SoTA Imagen 3 across general-purpose and skill-specific prompt sets. We empirically demonstrate how our method's superiority lies in its adaptability and how the optimal CFG values change depending on the requirements of the prompt.

## 2 RELATED WORK

**Evaluation of text-to-image models.** Evaluating the output of text-to-image models is a significant challenge in itself. Beyond traditional metrics, such as FID (Heusel et al., 2017) for image quality, and CLIPScore for alignment, that cannot offer fine-grained feedback on sample quality, recent work has developed VQA-based systems as autoraters (Wiles et al., 2024; Hu et al., 2023; Yarom et al., 2024; Lin et al., 2024). These autoraters show strong correlation with human perception, but their reliance on large language models (LLMs) makes them too computationally expensive for use *during* the iterative inference process, relegating them to post-hoc analysis. This motivates the search for evaluators that are both effective and efficient enough for online, step-by-step guidance. The most related work in this direction is that of Becker et al. (2025), Xu et al. (2023), Na

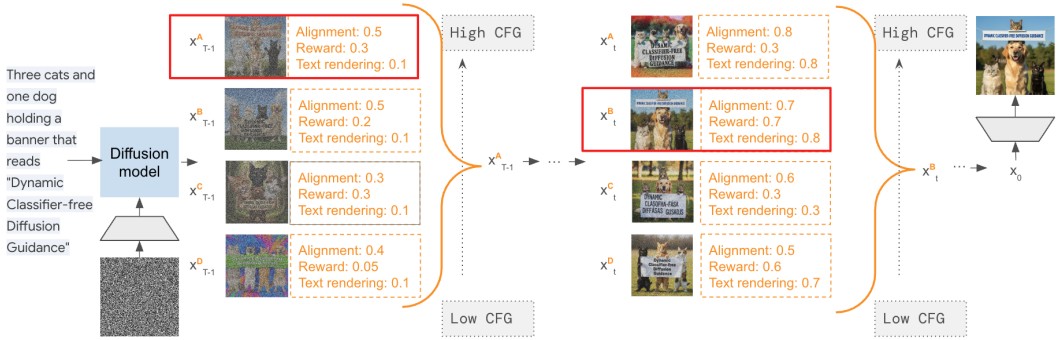

Figure 1: **Dynamic CFG**. We propose to perform a greedy search over multiple CFG scales and select the one that maximizes the latent evaluators' scores at each sampling step. The evaluators are small-scale and operate directly in the diffusion latent space increasing the computational overhead during inference by only 1%. Finally, for combining scores by multiple evaluators, we propose an adaptive weighting dependent on the denoising timestep.

et al. (2024), and Singhal et al. (2025). Becker et al. (2025) employ CLIP for evaluation directly in the latent space but they only assess denoised latents before the final decoding step. Xu et al. (2023) and Na et al. (2024) use a discriminator for evaluating visual quality during sampling for rejecting poor quality samples or restart the process earlier on. Finally, Singhal et al. (2025) and Kim et al. (2025) propose FK steering and DAS, respectively, for improving sampling starting from multiple random seeds and evaluating the intermediate "potentials" of samples. We introduce a flexible framework for combining feedback from multiple general and capability-specific evaluators to enable more fine-grained, multi-faceted control. Crucially, in contrast to prior work, we do not increase the NFEs and aim at improving a single seed instead of choosing or steering multiple seeds. Our method is orthogonal to work that rejects bad initial seeds.

**Guided image generation.** Classifier-free guidance (Ho & Salimans, 2022) has emerged as a useful way of trading-off sample quality and diversity using a single parameter. Recent work has focused on tuning the CFG values: Kynkäänniemi et al. (2024) apply guidance only for a limited time interval, and Chang et al. (2023) find that using a linearly increasing CFG schedule improves diversity. To improve sample quality and alignment, Sadat et al. (2023) use custom CFG schedules, while Wang et al. (2024) find that tuning such schedules per model and prompt set further improves results. In an attempt to correct for mistakes caused by CFG, Nichol et al. (2022) propose to additionally employ classifier guidance via a noise-conditioned CLIP model which gradients push samples towards the direction of the prompt. In the opposite end of the spectrum, Kim et al. (2023) propose a similar method using a discriminator for increasing visual fidelity. However, combining CFG with auxiliary model guidance increases complexity, makes manual hyperparameter tuning more strenuous and does not offer different guidance strength depending on the prompt.

## 3 METHOD

### 3.1 PRELIMINARIES

Diffusion models are a class of generative models that learn to reverse a noising process and are defined by two Markov processes. The forward process iteratively adds Gaussian noise to the data $x_0$ with $T$ increasingly noisy steps. At timestep $t \in [1, T]$ noise is added to $x_0$ as follows: $\mathbf{x}_t = \sqrt{\alpha_t}\mathbf{x}_0 + \sqrt{1 - \alpha_t}\boldsymbol{\epsilon}_t, \boldsymbol{\epsilon}_t \sim \mathcal{N}(\mathbf{0}, \mathbf{I})$, where $\alpha_t \in (0, 1)$ are pre-defined schedule parameters. The learned backward process gradually denoises $x_T$ towards the data distribution $p(x_{data})$. After training a diffusion model $p_\theta(x_0)$ to fit the data distribution, we sample from it starting with Gaussian noise: $\hat{\mathbf{x}}_0 = \frac{1}{\sqrt{\alpha_t}}\left(\mathbf{x}_t - \sqrt{1 - \alpha_t}\boldsymbol{\epsilon}_\theta(\mathbf{x}_t, t)\right)$, where $\boldsymbol{\epsilon}_\theta(\mathbf{x}_t, t)$ is the model's noise prediction.

## 3.2 ONLINE EVALUATORS

Given a noisy latent sample $x_t$ at denoising step $t$, we compute a score $e_t$ for evaluating the sample's quality across a specific dimension using one of the following evaluators.

**Alignment.** Given $x_t$ and the conditioning prompt $c$, we compute noisy latent CLIP scores as a prediction of final sample alignment:

$$e_{\text{CLIP}} = CLIP_{\text{vision}}x_t * CLIP_{\text{text}}c^T \tag{1}$$

CLIP is initialized from a standard pre-trained model trained on clean real images and corresponding captions from the WebLI dataset (Chen et al., 2023b). We replace the embedding layer of the vision encoder with a randomly initialized one matching the dimensionality of the diffusion encoder. We then fine-tune the model on image-text pairs after encoding the images into diffusion latents and injecting random noise with a similar time schedule as for the diffusion model training. We further condition the vision encoder on timestep $t$ converting CLIP into a time-conditioned encoder. We use the standard CLIP contrastive objective to map noisy latents to text descriptions.

**Visual quality.** Given $x_t$, we compute a score corresponding to the likelihood of an image being real independently of $c$ via a noisy latent Discriminator trained to differentiate between real and generated images, similar to prior work (Kim et al., 2023; Na et al., 2024):

$$e_{\text{Disc}} = -log\frac{p(x_t|t)}{1 - p(x_t|t)} \tag{2}$$

where $p(x_t|t)$ is the time-conditional probability of image $x_t$ to be real on timestep $t$. We initialize the discriminator from the latent CLIP vision encoder and introduce a classification head on top for predicting whether the images are synthetic or real. We train the discriminator on a small set of real vs. generated images from the MSCOCO dataset (Lin et al., 2014), similar to Kim et al. (2023).

**Reward (Human preference).** Similarly to reward modeling, we further fine-tune the latent alignment evaluator on pairs of generated images for the same prompt given human preference labels that reflect overall preference (aesthetics, alignment, artifacts). For converting pairwise comparisons to scores, we follow common approaches from LLM alignment (Ouyang et al., 2022) for reward tuning and use the Bradley-Terry (BT) model (Bradley & Terry, 1952). According to the BT model, CLIP is further optimized according to the following training objective:

$$p(i > j|c) = \frac{p(i|c)}{p(i|c) + p(j|c)} \tag{3}$$

where $p(i|c)$ and $p(j|c)$ is CLIP similarity between the prompt $c$ and each image $i, j$ in the comparison pair, with $i$ being the preferred one.

**Text rendering.** We consider a capability-specific evaluator for text rendering, a challenging aspect in image generation. We fine-tune the alignment evaluator on generated images labeled with scores by an OCR model. We introduce a multimodal head on top of the dual encoder and train the model to predict text rendering specific scores. We optimize the evaluator with an MSE objective:

$$\text{MSE}_{\text{TR}} = \frac{1}{n}\sum_{i=1}^{n}(e_{TR}^i - e_{OCR}^i)^2 \tag{4}$$

where $e_{TR}, e_{OCR}$ are the scores predicted by the latent evaluator and OCR model, respectively.

**Numerical Reasoning.** We consider another capability-specific evaluator for numerical reasoning by fine-tuning the noisy latent CLIP on a subset of WebLI-100B images (Wang et al., 2025) filtered to contain countable entities. We fine-tune the model with the original contrastive objective on the capability-specific dataset.

## 3.3 DYNAMIC CFG SEARCH VIA ONLINE FEEDBACK

**Dynamic CFG.** Classifier-free guidance (CFG) (Ho & Salimans, 2021) alleviates the need of a classifier for generating samples with high fidelity and mode coverage. In CFG, a model is trained to be both conditional and unconditional, and the respective scores are combined during generation via the CFG scale $s$, which regulates the trade-off between fidelity, alignment and diversity:

$$\epsilon_\theta(x_t|c) = \epsilon_\theta(x_t|\emptyset) + s(\epsilon_\theta(x_t|c) - \epsilon_\theta(x_t|\emptyset)) \tag{5}$$

where $\theta$ is the parameters of the diffusion model, $c$ is the condition applied to the diffusion model, i.e., the prompt for text-to-image generation, and $\emptyset$ is an empty sequence used for training the unconditional variant of the diffusion model.

We propose to dynamically select the optimal CFG scale *per timestep* given feedback $e$ from the online evaluators of Section 3.2 (see Figure 1). Formally, given a set of CFG scales $S = \{s_1, s_2, \ldots, s_n\}$, at every step we select the scale

$$\hat{s}_t = arg\,max_{s \in S} e_t(x_t^s, c), \tag{6}$$

which maximises the timestep-conditioned evaluator's score $e_t$ for the conditioning prompt $c$.

We optimize the final sample quality via a *greedy* search across timesteps, selecting the CFG scale that maximizes our latent evaluators' scores per step. Crucially, this search is performed without increasing the Number of Function Evaluations (NFEs). For each timestep $t$, we denoise once to obtain the conditional $\epsilon_\theta(x_t|c)$ and unconditional $\epsilon_\theta(x_t|\emptyset)$ predictions, and then cheaply test multiple CFG scales via Equation 5. Since our latent evaluators are lightweight and operate directly in the latent space, there is no increase in computation during inference (around 1% increase in FLOPs in contrast to 400% increase if operating in the pixel-space, see details in Appendix A.3). We further validate this design choice in Section 5.2, demonstrating that our greedy approach achieves performance comparable to computationally intensive beam search strategies.

**Adaptive evaluators' weighting.** We aim to combine feedback from general and capability-specific evaluators. Intuitively, our approach is founded on the principle that different properties emerge at different stages of generation. For example, coarse-grained alignment is established early on, while text legibility and artifact removal are late-stage concerns. Prior work also notes that high initial guidance can degrade visual quality (Wang et al., 2024). Given this sampling time-dependency, a static linear weighting of evaluator scores is insufficient. We therefore employ a dynamic weighting scheme that adjusts the influence of each evaluator $e \in E$ according to the current timestep, a strategy we show to be critical for optimal performance in Section 5.2.

$$\hat{e}_t = \sum_{e \in E} \alpha_{e,t} * e_t, \quad \text{where} \quad \alpha_{e,t} = \frac{e_t - e_{t+1}}{e_{t+1}}. \tag{7}$$

Intuitively, our dynamic weighting scheme amplifies an evaluator's influence at the precise moment its signal becomes meaningful, which we identify by detecting a significant change in its score across timesteps—a sign that the generation has entered an information-rich phase for that property.

## 4 EXPERIMENTAL SETUP

**Diffusion Models.** We experiment with both open-source and SoTA proprietary model families. We use **LDM** (i.e., latent diffusion model), a variant of the open-source StableDiffusion (Rombach et al., 2022) text-to-image model, trained on web-scale image data. We use $LDM_{small}$ (865M parameters) for ablations and $LDM_{large}$ (2B parameters) for main results. We also transfer our approach to **Imagen 3** (Team et al., 2024) and test whether our improvements hold on near-perfect text-to-image generation. For each model family we train separate evaluators tuned on the respective latent spaces.

**Prompt Sets.** We use general purpose and specialized prompt sets for evaluating image generation performance across different generation aspects. We use **Gecko** (Wiles et al., 2024) and **GenAI-Bench** (Li et al., 2024), which are diverse prompt sets containing fine-grained categories, for measuring overall preference in text-to-image generation. We use **MS-COCO** eval (Lin et al., 2014) for automatic evaluation on visual fidelity due to access to the ground-truth reference images, **MARIO-eval** (Chen et al., 2023a) for evaluating text rendering, and **GeckoNum** (Kajić et al., 2024) for testing numerical reasoning (i.e., counting).

**Evaluation.** For automatic evaluation, we use Gecko score (Wiles et al., 2024) for measuring fine-grained text alignment and FID (Heusel et al., 2017) on MS-COCO for measuring fidelity. For human evaluation, we run studies via side-by-side comparisons between model variants and report win rates over the baseline marking significance with 95% confidence intervals. For Gecko and GenAI-Bench we ask raters to indicate the image that they overall prefer (with respect to both alignment and aesthetics), for MARIO-eval we ask them to choose the image with the best aligned rendered text, and for GeckoNum we ask them to indicate the image that more closely represents the correct count of objects/entities (see details in Appendix A.4).

Table 1: **Filtering performance**. We evaluate the degree of prompt alignment via the Gecko score while filtering samples of poor alignment at different % during sampling. For filtering, we either use the latent CLIP evaluator or an off-the-shelf CLIP model operating in the pixel space. In all cases, we select the best out of a batch of 4 when filtering. Computed on the Gecko prompt set.

| Model | Evaluator | No filtering | Filter @ [Gecko Score] | | | |
| --- | --- | --- | --- | --- | --- | --- |
| | | | 25% | 50% | 75% | 100% |
| $LDM_{small}$ | latent-space CLIP | 37.6 | 39.7 | 41.4 | 43.0 | 43.0 |
| | pixel-space CLIP | 37.6 | 43.4 | 44.6 | 44.7 | 45.1 |
| $LDM_{large}$ | latent-space CLIP | 42.9 | 45.9 | 45.2 | 46.6 | 46.0 |
| | pixel-space CLIP | 42.9 | 47.1 | 48.9 | 48.4 | 48.6 |

**Latent evaluators' training.**    Our analysis reveals that the reliability of feedback from our latent evaluators depends heavily on the noise level. While coarse attributes like overall visual structure and semantic alignment can be assessed early in generation, fine-grained details—such as minor artifacts or the legibility of rendered text—can only be evaluated accurately at lower noise levels. This motivates a time-weighted loss schedule for the human feedback and text rendering evaluators. We provide details on training and computational requirements in Appendix A.1.

## 5    RESULTS

### 5.1    EVALUATION OF LATENT EVALUATORS

We evaluate the effectiveness of the latent evaluators described in Section 3.2 by answering two questions: 1. What is the information loss by directly assessing compressed latents instead of pixel-space images? 2. How early during denoising can we get signal for sample quality?

Similarly to Karthik et al. (2023) and Astolfi et al. (2024), we perform filtering for evaluating the effectiveness of the evaluators. Instead of filtering samples after denoising, we evaluate potential paths during generation. We consider a large number $B$ of initial seeds per prompt and aim at subselecting the $K$ best ones at timestep $t$. We explore filtering at different timesteps $t$ corresponding to a different percentage of NFEs.

We report the Gecko score on $LDM_{small}$/$LDM_{large}$ when filtering images via the alignment (CLIP) evaluator at different sampling stages in Table 1. We compare the performance of the latent evaluator against a pixel-space equivalent. In this case, we first perform one-step denoising from $x_t$ to $x_0$ and decoding of $x_0$ into pixels, which produces clean but blurry images that can be processed by an off-the-shelf encoder. We find that the information loss we suffer by operating directly on latents is consistent for different noise levels. Although there is an expected performance drop when using latents, we still maintain information about sample quality while reducing the computational overhead allowing us to use the latent evaluators online during inference (see Appendix A.3). Importantly, we find that we correctly discard poorly aligned samples from as early as 25% of the denoising process. We observe a similar behavior for the visual quality evaluator (see Appendix A.5).

### 5.2    DYNAMIC CFG SEARCH

**LDM.**    We compare our dynamic CFG search against gradient-based guidance (Nichol et al., 2022; Kim et al., 2023) and static CFG schedules (Kynkäänniemi et al., 2024; Sadat et al., 2023) on $LDM_{large}$ in Table 2 using the automatic metrics described in Section 4.

Alignment (CLIP) guidance is indeed effective for improving alignment without any benefits in visual fidelity, whereas the visual quality (Discriminator) guidance only slightly improves alignment, but not FID. When combining the gradients of the two models, we observe no effect; while CLIP improves alignment, discriminator guidance fails to boost fidelity. In contrast, our dynamic CFG search (last block of Table 2) demonstrates a clear and controllable trade-off. Using only the alignment evaluator optimizes the Gecko score, while using only the visual quality evaluator optimizes FID. Our full approach leveraging adaptive weighting to combine the evaluators, successfully

Table 2: **Automatic evaluation on LDM$_{\text{large}}$**. We report alignment and visual fidelity performance via Gecko score and FID respectively for (1) gradient-based guidance that uses auxiliary models for correcting samples, (2) static CFG schedules derived from empirical observations, and (3) our dynamic CFG search when using latent alignment and/or visual quality (VQ) evaluators.

| Method | Latent evaluator/ Static schedule | Gecko score ↑ (Gecko prompts) | FID ↓ (MS COCO prompts) |
|---|---|---|---|
| Default CFG (fixed) | – | 43.8 | 25.6 |
| Gradient guidance | Alignment (Nichol et al., 2022) | 46.1 | 25.6 |
| | VQ (Kim et al., 2023) | 44.6 | 25.5 |
| | Alignment + VQ | 45.3 | 25.5 |
| Static CFG schedules | Limited Guidance Interval (Kynkäänniemi et al., 2024) | 43.0 | 26.1 |
| | Annealing (Sadat et al., 2023) | 47.0 | 28.9 |
| | Mean of Dynamic CFG | 46.5 | 26.8 |
| | Median of Dynamic CFG | 45.8 | 26.0 |
| Dynamic CFG search | Alignment | 45.5 | 26.4 |
| | VQ | 44.0 | **24.8** |
| | Alignment + VQ (linear) | 45.0 | 25.4 |
| | Alignment + VQ (adaptive) | **47.2** | **24.8** |

improves both dimensions at once. We find the adaptive weighting to be critical: using a static, time-independent weighting significantly hurts performance.

We first compare the dynamic CFG search against a constant value, limited-interval guidance (Kynkäänniemi et al., 2024) and an annealing schedule (Sadat et al., 2023) (third block of Table 2). While the annealing schedule improves alignment at the cost of visual fidelity, our dynamic schedule matches its alignment performance while simultaneously improving fidelity. To determine if the gain comes from the schedule's general shape or its per-prompt adaptability, we create a static "mean schedule" by averaging our dynamic schedules over all prompts and apply it universally. We find that performance drops in this condition, which, while still competitive, highlights that the per-prompt adaptability of our approach is a crucial component of our method's success.

**Imagen 3.** We next assess how our method transfers to Imagen 3 via human evaluation as described in Section 4. We extend the suite of latent evaluators since we find the discriminator to be an insufficient visual quality predictor for Imagen 3 during early experimentation[1]. As discussed in Section 3.2, we instead use a reward evaluator trained on human preference data alongside with two capability-specific evaluators: one for text rendering and one for numerical reasoning.

We report win rates of side-by-side comparisons in Table 3 across Gecko, GenAI-Bench, MARIO-eval and GeckoNum. Our dynamic CFG framework yields statistically significant improvements over the strong Imagen 3 baseline. Consistent with our findings on LDM, using either the alignment or the reward evaluator is preferred over the baseline across all prompt sets. We further validate that combining the two evaluators with adaptive weighting achieves the best results across all prompt sets reaching up to 54.7% win rate on MARIO-eval for text rendering.

We demonstrate the flexibility of our framework by also deploying two specialized evaluators for text rendering and numerical reasoning. We test their effectiveness on specialized prompt sets tailored for measuring each capability separately. On these prompt sets, we find that both evaluators achieve the highest win rates against the baseline (55.5% on text rendering and 54.1% on numerical reasoning) when also combined adaptively with the general purpose evaluators (either alignment or reward).

We additionally report the performance of the heuristic CFG schedules (Kynkäänniemi et al., 2024; Sadat et al., 2023) as applied in LDM on Imagen 3. The results are striking: the schedules that offered modest improvements on LDM fail on Imagen 3, degrading performance below the baseline in most cases. This failure underscores a fundamental weakness of heuristic-based methods: they are brittle because they rely on empirical rules derived from a specific model architecture and training

---

[1]We hypothesize that since Imagen can generate very high quality photorealistic images, predicting small artifacts or aesthetic improvements via a discriminator can be more challenging than on LDM.

Table 3: **Human Preference on Imagen 3**. Side-by-side human comparisons of the baseline Imagen 3 and Imagen 3 with our dynamic CFG search. We report win rates for the custom CFG schedules against the default and underline the wins that are significant with a 95% confidence interval. We report results on Gecko and GenAI-Bench for overall preference, MARIO-eval for text rendering and GeckoNum for numerical reasoning.

| Method | Latent Evaluator | Win Rate (%) ↑ | | | |
| --- | --- | --- | --- | --- | --- |
| | | Gecko | GenAI-Bench | MARIO-eval | GeckoNum |
| Limited Interval | – | 27.9 | 33.1 | 19.6 | 46.6 |
| Annealing | – | 46.4 | 34.4 | 42.7 | 50.8 |
| | Alignment | 50.9 | 53.2 | 52.3 | 51.1 |
| Dynamic CFG | Reward | 52.1 | 51.4 | 53.8 | 53.8 |
| | Alignment + Reward | **53.6** | **53.8** | 54.7 | 53.6 |
| *Capability-specific evaluators* | | | | | |
| | Text rendering | – | – | 53.1 | – |
| | + Alignment | – | – | **55.3** | – |
| | + Reward | – | – | **55.5** | – |
| Dynamic CFG | Numerical | – | – | – | 52.2 |
| | + Alignment | – | – | – | 53.2 |
| | + Reward | – | – | – | **54.1** |

regime. When exploring the interval-based guidance in particular, we find that this schedule fails completely for text rendering specific prompts. This agrees with our intuition that text rendering benefits from higher guidance throughout, but also in the final sampling timesteps which the prompt independent schedules do not take into account. In contrast, both heuristic schedules perform best on prompts related to numerical reasoning indicating that lower guidance strength in the beginning of denoising favors diversity for producing entities and objects in variable numbers. Our method's strength lies in its model-agnostic, online adaptation. Instead of applying a pre-determined, "hard-coded" schedule, derived after cumbersome hyper-parameter search, our framework discovers the optimal guidance on-the-fly by reacting directly to the outputs of the target model. This is why our approach generalizes out-of-the-box from a weaker to a state-of-the-art model and consistently improves performance across different generation skills.

**Dynamic CFG vs FK steering** We finally compare our method against FK steering Singhal et al. (2025) on the Gecko prompt set to evaluate the trade-off between computational cost and generation quality. While FK steering involves searching over multiple initial seeds—significantly increasing NFEs and inference cost—our Dynamic CFG operates on a fixed seed with only a ∼1% computational overhead. We consider FK steering with $k = 4$ particles for Imagen 3 and conduct a side-by-side human evaluation against our method when using the general-purpose (alignment and reward) latent evaluators. The analysis results in a win rate of 50.7% for FK steering (95% CI $[48.9\%, 52.5\%]$) against our approach. This confidence interval implies the two methods are statistically indistinguishable in terms of overall human preference and the results demonstrate that Dynamic CFG achieves performance comparable to compute-intensive inference-time scaling strategies without the associated increase in NFEs or latency.

**Greedy CFG Scale Selection vs. Global Optimization.** While our method utilizes a greedy selection strategy to determine the optimal CFG scale at each timestep, we acknowledge that this does not theoretically guarantee a global maximum for the final sample quality. However, the stability of our latent evaluators supports this approximation; as detailed in Section 5.1, these evaluators are robust predictors of final image alignment, maintaining consistent predictions even during early denoising stages (e.g., from 25% completion). This approach parallels greedy decoding strategies commonly used in LLM inference, which prioritize efficiency and provide strong approximate solutions without the exhaustive cost of global search.

To empirically validate the effectiveness of our greedy approach against a more exhaustive search, we implement a beam search strategy. At each denoising step $t$, instead of selecting a single optimal trajectory, this method maintains the top-$k$ candidates. For each candidate $i$, we generate potential

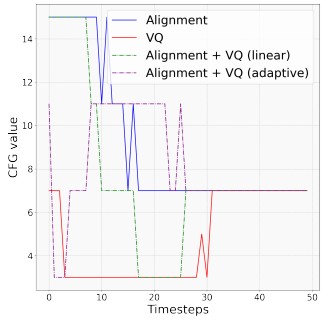 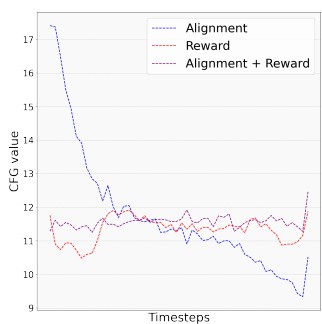 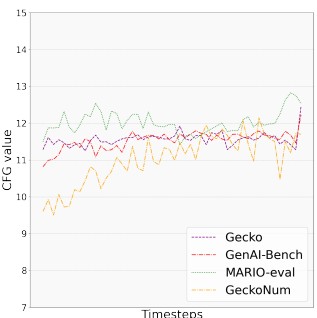

(a) Median values in LDM for the Gecko prompt set when using an alignment (CLIP) or visual quality (VQ) evaluator or their combination with a fixed linear or adaptive weighting.

(b) Smoothed median normalized values in Imagen 3 for the Gecko prompt set when using an alignment (CLIP) or reward (Human pref) evaluator or their combination with adaptive weighting.

(c) Smoothed median normalized values in Imagen 3 for the different prompts when using the best performing combination of evaluators as shown in Table 3.

Figure 2: Median of the dynamic CFG schedule on different models and prompt sets.

**Low Guidance in Dynamic CFG** · · · · · · · · · · **High Guidance in Dynamic CFG**

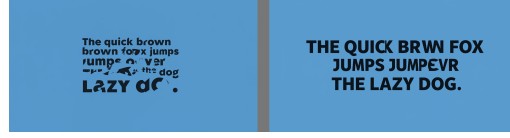

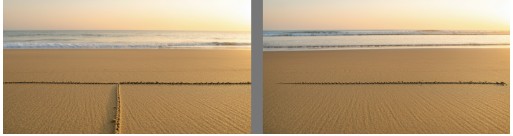

(a) Prompt: "...pop art depicting the Mona Lisa... **blocks of bright pink and yellow in a checkered design**, with **a touch of orange and white**..."

(b) Prompt: "A photograph of a **thin, white line drawn in the sand** on a beach at sunrise. **The line is straight, clean and simple**..."

(c) Prompt: "**The quick brown fox jumps over the lazy dog**, written in serif font. "

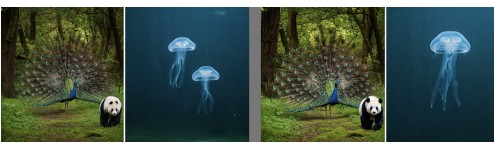

(d) Prompt: "A peacock fans it's plumage while a panda is walking and **a jellyfish is swimming in the ocean**."

Figure 3: We rank images by Imagen 3 from lowest to highest guidance strength when using dynamic CFG for the Gecko prompt set. For each prompt, we present a pair of images for **default (left) vs dynamic CFG (right)**. Validating our hypothesis, creative or simple prompts get low guidance, whereas prompts including text rendering and compositionality get the highest guidance.

next states $x_{t-1}^i$ across a range of CFG values and prune the set to retain only the top-$k$ paths for the subsequent step.

The primary constraint of beam search is the computational penalty; specifically, a beam width of $k$ scales the NFEs by a factor of $k$. We compare our standard greedy approach against a beam search with $k = 2$ (doubling the inference cost) on Imagen 3 using the Gecko prompt set. Utilizing the latent CLIP evaluator for guidance and the Gecko metric for final quality assessment, we observe that beam search yields a marginal improvement, achieving a score of 73.7% compared to 73.1% for our greedy method. Given that the 0.6% performance gain requires a 100% increase in NFEs, we conclude that greedy selection offers the superior trade-off between generation quality and computational efficiency.

## 5.3 DYNAMIC CFG SCHEDULE

**LDM.** Figure 2a visualizes the median CFG schedule on LDM$_{\text{large}}$. The behavior of the individual evaluators confirms they are working as intended, defining the extremes of the alignment-fidelity

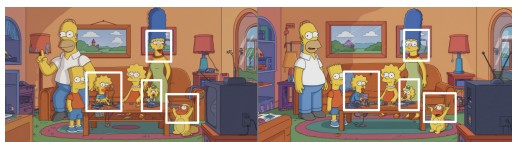
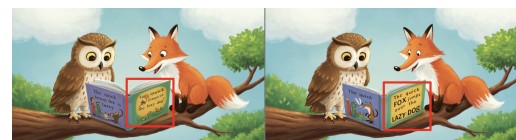

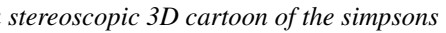

*a stereoscopic 3D cartoon of the simpsons*

*"The quick brown fox jumps over the lazy dog"*

(a) Artifact correction

(b) Text Rendering

Figure 4: Qualitative examples for Imagen 3 on the Gecko prompt set when using default sampling (left) vs our dynamic search (right).

trade-off. The alignment evaluator consistently favors high CFG scales to maximize alignment, while the visual quality one pushes towards low scales (approaching unconditional generation) to maximize fidelity. Our full method, using adaptive weighting, successfully navigates this trade-off. It generates an arc-shaped schedule that avoids extreme CFG values at the beginning and end of sampling. This emergent shape aligns with empirical findings from prior work (Wang et al., 2024). In contrast, a static weighting of the evaluators fails to find this balance and produces a schedule largely dominated by the alignment signal.

**Imagen 3.** We present the smoothed normalized median of the dynamic CFG schedule for Imagen 3 in Figure 2b when using either of the alignment or reward evaluators or their combination. Similarly to LDM the alignment evaluator favors high guidance strength in the beginning of denoising, but the optimal median schedule derived by the combination of the two evaluators significantly differs from the one discovered for LDM. This further validates that no empirical observations regarding CFG can generalize beyond a specific model family, highlighting the strength of our dynamic approach that can adapt to different models consistently providing improvements.

We also present the smoothed normalized CFG schedule for the best performing variant of our dynamic CFG per prompt set in Figure 2c. We find that the patterns in the CFG schedules agree with our empirical observations: in contrast to the general-purpose prompt sets, text rendering (MARIO-eval) on average requires higher guidance strength especially in the end of denoising, and numerical reasoning (GeckoNum) benefits from lower guidance strength in the beginning of generation which favors diversity and avoids "template-like" generations of objects and entities allowing the model to generalize to variable counts. We further rank the generated images for the Gecko prompt set, which contains diverse prompt categories, based on the average selected CFG across timesteps when using dynamic CFG. We present in Figure 3 two of the lowest ranking examples on the left (i.e., low guidance strength) and two of the highest ranking ones. The visualization further validates our hypothesis that the degree of guidance is dependent on the requirements of the prompt. Indeed, creative or simple prompts benefit from low CFG values, whereas prompts that require strong alignment, such as text rendering and compositionality, need much higher guidance strength. We present additional qualitative results in Appendix A.6.

## 6    CONCLUSIONS

In this paper, we propose a framework for dynamically selecting the optimal CFG scale during denoising in text-to-image generation. We demonstrate that the optimal trade-off between conditional and unconditional generation is not fixed, but rather a dynamic function of the prompts' content, the sampling timestep, and the diffusion model. We suggest a suite of latent evaluators for assessing both general purpose (alignment, visual quality) and specialized (text rendering, numerical reasoning) properties of generation and demonstrate that we can successfully use them *during* diffusion inference at minimal computational cost. Crucially, while our method relies on the quality of these evaluators, we find that even general-purpose models trained on noisy data generalize effectively to diverse tasks without specialized fine-tuning. Furthermore, performance can be maximized by curating domain-specific training data, offering a flexible pathway for future enhancements. Given such evaluators, our proposed dynamic CFG significantly boosts generation quality on both weaker (gLDM) and more powerful (Imagen) models, validating the generalization of the approach. Our approach can be extended to more specialized skills given appropriate evaluators and the framework can be expanded to perform inference-time search beyond the CFG schedule.

**Acknowledgements**   We thank Benigno Uria, Oliver Wang, and Zhisheng Xiao for their feedback throughout the project. We are grateful to Chris Dyer and Sayna Ebrahimi for their feedback on the manuscript and Jordi Pont-Tuset for his help in human evaluation.

**Ethics Statement**   The full details of the human evaluation study design presented in Section 5.2 and Table 3, including compensation rates, were reviewed by our institution's independent ethical review committee. All participants provided informed consent prior to completing tasks and were reimbursed for their time.

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

## A    APPENDIX

### A.1    LATENT EVALUATORS

**Training.** We initialize the latent alignment (CLIP) evaluator with a pre-trained CLIP model trained on the WebLI dataset. We use a pre-trained CLIP-ViT-B/16 (Radford et al., 2021; Zhai et al., 2023) model version with a ViT-B vision encoder and a BERT-Base (Devlin et al., 2019) text encoder. The dual encoder has in total 194M parameters.

As mentioned in Section 3.2, we randomly initialize the embedding layer of the vision encoder in order to change the pixel-space embedding layer to a diffusion-specific latent-space one. Specifically, for LDM we convert ViT-B/16 to ViT-B/4 resulting in a 256 token sequence for an image with initial resolution of (512, 512) encoded into latents. Accordingly, we also change the embedding layer for Imagen 3. We then fine-tune the whole model on noisy diffusion latents encoded and corrupted from image-text papers of the WebLI dataset. We fine-tune the model for 90k steps using a batch size of 512. We use a cosine learning rate schedule with linear warm up and no weight decay. Our base learning rate is $5e^{-5}$. We train our model on 64 TPUv5e chips for 1.5 days.

We initialize all other latent evaluators with the above latent alignment evaluator and continue fine-tuning the whole network for approximately 10k steps on the capability-specific data as described in Section 3.2 and summarized in Table 4.

Table 4: Training data per latent evaluator.

| Latent evaluator | Training data |
| --- | --- |
| Alignment evaluator | WebLI (Chen et al., 2023b) |
| Visual quality evaluator | Real & generated images from MSCOCO (Lin et al., 2014) |
| Reward evaluator | Human preference data on generated images |
| Text rendering | OCR scores on generated images |
| Numerical reasoning | 100K re-captioned image-text pairs by Gemini 2.5 Pro for accurate descriptions of object counts |

We observe that for the reward and text rendering evaluators, which measure fine-grained qualities in image generation, a useful signal only emerges for timesteps $t < t_{min} + \frac{1}{3}(t_{max} - t_{min})$. Consequently, during the initial high-noise phase of generation ($t > t_{min} + \frac{1}{3}(t_{max} - t_{min})$), we apply a

near-zero weight to their corresponding loss. For the subsequent phase ($t < t_{min} + \frac{1}{3}(t_{max} - t_{min})$), as the noise level decreases, we increase the loss weight. We experiment with schedules where this weight ramps up—either linearly or exponentially—from its initial low value, reaching a maximum of 1 at the final timestep ($t = t_{min}$):

$$w_{loss}(t) = \begin{cases} 0.05 & \text{if } t > t_{min} + \frac{1}{3}(t_{max} - t_{min}) \\ 0.05 + 0.95 \cdot \frac{e^{\frac{k(t-\alpha)}{\beta}} - 1}{e^k - 1} & \text{otherwise} \end{cases} \tag{8}$$

where $t_{max}$ is the timestep corresponding to pure noise, $t_{min}$ corresponds to clean data, $\alpha = \frac{2(t_{max} - t_{min})}{3}$, $\beta = \frac{t_{max} + 2t_{min}}{3}$ and $k$ is a hyper-parameter defining the sharpness of the curve which we set to 5.

## A.2 DYNAMIC CFG SEARCH

**CFG values.** We find that the best default (fixed) value for both $LDM_{small}$ and $LDM_{large}$ is 7.5. For our dynamic CFG search, we are searching over the following set of 5 CFG values: $[1, 3, 7.5, 11, 15]$ for all denoising timesteps. For Imagen 3, we extend our search to a set of 24 discrete CFG values.

## A.3 COMPUTE

We report FLOPs for different model functions (i.e., denoising, decoding, online evaluation) and for the full denoising process for the LDM model in Table 5.

We overall use evaluators that are small and lightweight in order to be computationally efficient in our online sampling setting. By operating in the latent space directly we use a latent CLIP model which is 4 times more efficient than the pixel-space equivalent due to the compressed inputs. Crucially, when using a latent evaluator, we do not require decoding the latents via the VAE at each denoising step. This reduces the computational cost from 4 times more than the baseline for the pixel-space evaluator, which is prohibited, to only 1% of the overall computation required for sampling from $LDM_{large}$.

Table 5: Comparison of FLOPS per model function.

| Model | FLOPS $\times 10^9$ |
|---|---|
| $LDM_{small}$ denoising step | 875 |
| $LDM_{large}$ denoising step | 2280 |
| VAE-decode | 1489 |
| Latent alignment evaluator | 5 |
| Pixel-space alignment evaluator | 22 |
| $LDM_{large}$: baseline sampling | 115,489 |
| $LDM_{large}$: sampling with latent evaluator | 116,739 |
| $LDM_{large}$: sampling with pixel-space evaluator | 493,239 |

## A.4 HUMAN EVALUATION

We recruited participants (N = 60) through an internal crowdsourcing pool. The full details of our study design, including compensation rates, were reviewed by our institution's independent ethical review committee. All participants provided informed consent prior to completing tasks and were reimbursed for their time. We collect and aggregate on average two to three ratings per prompt-image pair, considering both the wins of each model and the ties in the ratings.

For the Gecko and GenAI-Bench prompt sets, we display generated images by different model variants side-by-side for the same prompt and ask raters to indicate which one they overall prefer in terms of both aesthetics and prompt adherence (the options are to indicate one or none of the images). For the MARIO-eval prompt set, we again display the generated images side-by-side asking the raters to indicate the one they prefer in terms of text rendering, i.e., which one better

Table 6: **Filtering performance**. We report FID while filtering samples of poor visual quality at different % during sampling. For filtering, we use the visual quality evaluator and select the best out of a batch of 4 when filtering. Computed on the MS COCO prompt set.

| Model | Noisy evaluator | Baseline | Filter @ [FID ↓] | | | |
|---|---|---|---|---|---|---|
| | | | 25% | 50% | 75% | 100% |
| gLDM$_{large}$ | latent Disc | 29.2 | 27.6 | 27.4 | 27.0 | 26.8 |

visualizes the text requested by the prompt. Finally, for GeckoNum, we ask the raters to indicate the generated image out of the two that better reflects the number of objects or entities described in the prompt.

## A.5 ADDITIONAL EXPERIMENTAL RESULTS

**Evaluation of Latent Evaluators.** Additional to the results of Table 1 when using the alignment evaluator, we report the filtering performance of the latent visual quality evaluator on LDM in terms of FID on the Gecko prompt set in Table 6. We validate that the latent visual quality can correctly predict bad samples from as early as 25% offering improvements over the baseline.

**CFG Greedy Selection vs. Global Optimization.** While our method utilizes a greedy selection strategy to determine the optimal CFG scale at each timestep, we acknowledge that this does not theoretically guarantee a global maximum for the final sample quality. However, the stability of our latent evaluators supports this approximation; as detailed in Section 5.1, these evaluators are robust predictors of final image alignment, maintaining consistent predictions even during early denoising stages (e.g., from 25% completion). This approach parallels greedy decoding strategies commonly used in LLM inference, which prioritize efficiency and provide strong approximate solutions without the exhaustive cost of global search.

To empirically validate the effectiveness of our greedy approach against a more exhaustive search, we implement a beam search strategy. At each denoising step $t$, instead of selecting a single optimal trajectory, this method maintains the top-$k$ candidates. For each candidate $i$, we generate potential next states $x_{t-1}^i$ across a range of CFG values and prune the set to retain only the top-$k$ paths for the subsequent step.

The primary constraint of beam search is the computational penalty; specifically, a beam width of $k$ scales the NFEs by a factor of $k$. We compare our standard greedy approach against a beam search with $k = 2$ (doubling the inference cost) on Imagen 3 using the Gecko prompt set. Utilizing the latent CLIP evaluator for guidance and the Gecko metric for final quality assessment, we observe that beam search yields a marginal improvement, achieving a score of 73.7% compared to 73.1% for our greedy method. Given that the 0.6% performance gain requires a 100% increase in NFEs, we conclude that greedy selection offers the superior trade-off between generation quality and computational efficiency.

## A.6 QUALITATIVE EXAMPLES

**Qualitative Analysis on LDM.** Figure 5 provides a qualitative comparison between the default CFG and our dynamic approach on LDM, showcasing the effects of each latent evaluator. As the examples illustrate, the individual evaluators successfully target their respective domains but introduce trade-offs. Guiding with the discriminator alone enhances photorealism—for instance, improving the panda's fur texture in Example 2—but does so at the expense of prompt alignment, causing the koala from the prompt to disappear. Conversely, using only the CLIP evaluator enforces stronger prompt adherence, correctly adding glasses to the tiger in Example 1, but often at the cost of image quality and coherence, resulting in a "pasted-together" artifact. Our full method with adaptive weighting successfully resolves this tension, synthesizing the strengths of both evaluators to produce images that are both photorealistic and faithful to the prompt.

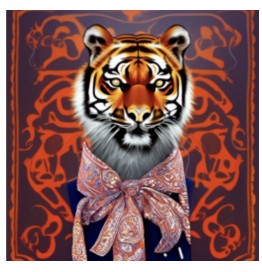 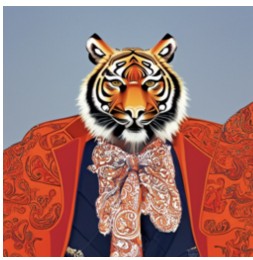 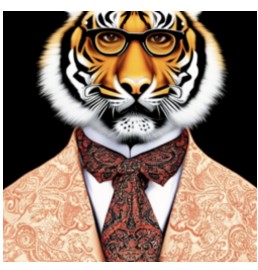 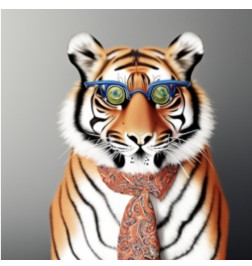

(a) Default CFG.      (b) Dynamic CFG (Disc).   (c) Dynamic CFG (CLIP).   (d) Dynamic CFG (CLIP + Disc).

*Prompt: "the tiger wears glasses and wears a paisley tie"*

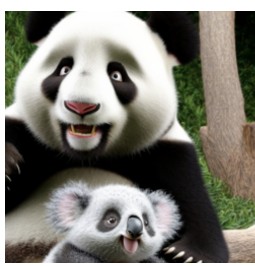 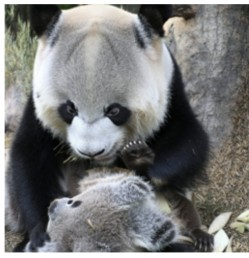 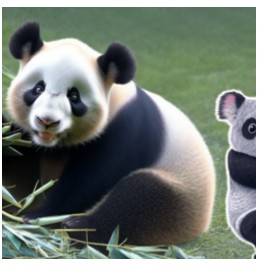 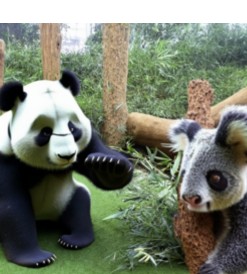

(e) Default CFG.      (f) Dynamic CFG (Disc).   (g) Dynamic CFG (CLIP).   (h) Dynamic CFG (CLIP + Disc).

*Prompt: "the panda waves to the koala bear"*

Figure 5: Qualitative examples for LDM when using different CFG schedules on the Gecko prompt set. The images of the first row are generated for the prompt: "*the tiger wears glasses and wears a paisley tie*" and the images of the second row are generated for the prompt: "*the panda waves to the koala bear*".

**Qualitative Improvements on Imagen 3.**   Next, in Figure 6, we demonstrate our method's ability to improve upon the already powerful Imagen 3 baseline. The qualitative improvements are most striking in areas where even state-of-the-art models can falter. Our dynamic CFG approach consistently reduces subtle visual artifacts, improves overall text alignment and, most notably, produces significantly more coherent and legible rendered text than the default sampler. This highlights our method's value not only for enhancing general quality but also as a tool for targeted improvements on specific, challenging generation tasks.

## A.7   LLM Use Disclosure

An LLM was used for polish writing of the paper and improving the phrasing of certain sentences. No LLM was used to write extended parts of the paper from scratch, or for retrieval, discovery and research ideation.

*Arifacts (Gecko).*

(a) Prompt: "An orange is being squashed under a glass bottle which is splintering into bits."

*Text alignment (GenAI-Bench).*

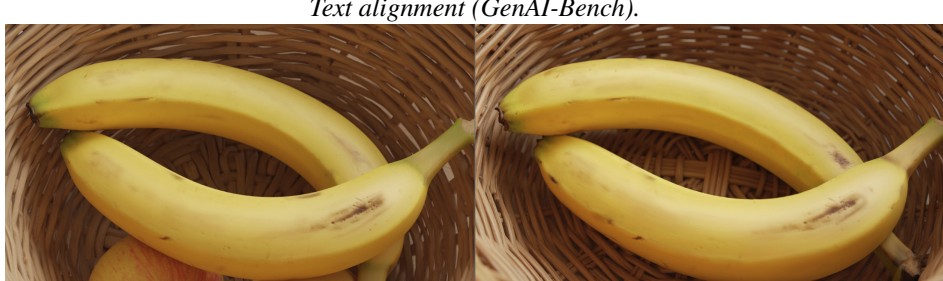

(b) Prompt: "There are two bananas in the basket, but no apples."

*Text rendering (MARIO-eval).*

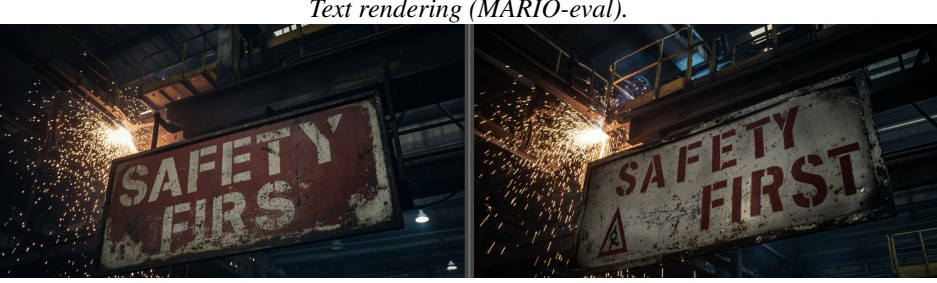

(c) Prompt: "In the factory, a sign that reads "Safety First"."

*Numerical reasoning (GeckoNum).*

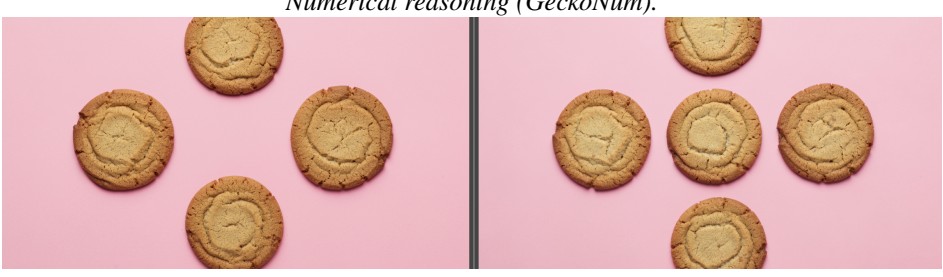

(d) Prompt: "5 cookies."

Figure 6: Qualitative examples for Imagen 3 on the Gecko prompt set when using different CFG schedules: default (left) vs ours dynamic (right). We observe improvements in alignment, artifacts, text rendering, and numerical reasoning.

