# OpenReview forum: "Dynamic Classifier-Free Diffusion Guidance via Online Feedback"
_ICLR.cc/2026/Conference — ICLR 2026 Poster_

### Official Review · Reviewer_Es1L · 2025-10-31

**Soundness:** 3
**Presentation:** 3
**Contribution:** 3
**Rating:** 8
**Confidence:** 4

**Summary:**

The paper introduces a new method to **dynamically regulate** the classifier-free guidance (**CFG**) scale in text-to-image diffusion models. At each denoising time step a guidance scale is greedily selected among a predefined set based on the feedback scores coming from different **evaluators**. These evaluators take in input a time step $t$ and a latent $x_t$ and evaluate: (a) the alignment to the text prompt, (b) the visual quality of the output, (c) the alignment w.r.t. human preferences, and optionally the quality of (d) text rendering or (e) numerical reasoning. The method is evaluated by experimenting different base models (LDM and Imagen 3) and different text prompt datasets (Gecko, GenAI-Bench, MARIO, GeckoNum). Both quantitative and qualitative results overall prove the effectiveness of the method and of each evaluator.

**Strengths:**

1. **Interesting use of the CFG scale.** I find the idea of using CFG scale to improve the alignment and the quality of text-to-image generative models very interesting and not too explored in the research community. I think the paper would definitely provide value to the ICLR community and inspire similar research directions.
2. **Interesting experiments.** In addition to the main contribution, I find the experiments reported in Figure 2 very interesting as they show how different CFG scales influences different metrics at different denoising time steps.
3. **Clear and well-structured presentation.** The paper is well structured and easy to follow.

**Weaknesses:**

1. **Ambiguity in algorithm description.** Lines 216–220 could be misread as computing all CFG outputs in parallel and assembling them afterward. Clarifying that the selection at each step influences subsequent sampling would be helpful.
2. **Partial evaluation of latent evaluators.** Section 5.1 reports detailed analysis only for the CLIP evaluator; others (e.g. discriminator) are not explicitly benchmarked.
3. **Model-agnostic claim.** I do not necessarily agree with the claim that the method is *model-agnostic* (line 428). Similarly to how baselines which rely on heuristics have to run hyper-parameter search to find the optimal CFG scale for a specific model, the proposed method also has to train the evaluators on the noisy latents coming from each specific model.

**Questions:**

Following the weakness listed above, I would like to ask these questions:
1. Lines 216–220 could be misread as computing all CFG outputs in parallel and assembling them afterward. Could this part be explained better?
2. Section 5.1 is named *Evaluation of latent evaluators*, but in practice only evaluates the performances of the latent CLIP evaluator and not of the others. Would it be possible to provide a justification of this choice?
3. Some minor issues:
    - Equation 6 refers to $W$ which, if I am not wrong, is never defined before. May this be $S$?
    - In Equation 7 the alpha coefficients could potentially assume negative values. Should there be an absolute value around the difference between $e_{t}$ and $e_{t+1}$?

---

> ### Author Response · Authors · 2025-11-20
> **Weakness Point 1**
>
> _Ambiguity in algorithm description. Lines 216–220 could be misread as computing all CFG outputs in parallel and assembling them afterward. Clarifying that the selection at each step influences subsequent sampling would be helpful._
>
> We have updated the manuscript to better explain how we search cheaply over different CFG values. Our initial description refers to **a single denoising step**: for timestep t, we denoise once to obtain the conditional (xt|c) and unconditional (xt|\empty) predictions. We then recombine the two predictions via Equation 5 considering different CFG scales, which does not require extra NFEs and it is cheap to do. The selection of the CFG scale at timestep t will influence the subsequent xt-1 since this will be the new input to the diffusion model.

---

> > ### Author Response · Authors · 2025-11-20
> > **Weakness Point 2**
> >
> > _Partial evaluation of latent evaluators. Section 5.1 reports detailed analysis only for the CLIP evaluator; others (e.g. discriminator) are not explicitly benchmarked._
> >
> > We also report **results for the discriminator that measures visual quality in Appendix A.5**. As we mention in Section 5.1, we observe a similar behavior for the discriminator, but we omitted the results from the main paper due to space limitations. Upon acceptance and when we will be given one extra page, we can transfer the results in the main paper and extend our discussion.

---

> ### Author Response · Authors · 2025-11-20
> **Weakness Point 3**
>
> _Model-agnostic claim. I do not necessarily agree with the claim that the method is model-agnostic (line 428). Similarly to how baselines which rely on heuristics have to run hyper-parameter search to find the optimal CFG scale for a specific model, the proposed method also has to train the evaluators on the noisy latents coming from each specific model._
>
> The latent evaluators depend on the diffusion latent space, so indeed they need to be retrained for different diffusion autoencoders. From this aspect, **the evaluators are not model-agnostic**. However, we show in the paper that **our framework is model-agnostic**, as our results transfer from a smaller open source latent diffusion model to SoTA image generation after post-training. What is interesting is that our framework can automatically find different patterns of CFG schedules for different model families: as shown in Figure 2, for LDM our framework proposes dynamic CFG schedules that on average agree with empirical observations on best CFG values for Stable Diffusion models. However, using the same exact framework and algorithm while adapting the evaluators, we see that the proposed dynamic CFG schedules for Imagen 3 are on average very different from our LDM results and empirical observations from the literature. Hence, we conclude that prior work that analyses the CFG schedules and claims agnostic patterns in generation cannot generalize to different model families. In contrast, our framework can retrieve optimal CFG schedules for different model families (given of course the fine-tuned evaluators that match the latent space).
>
> Regarding the comparison between running hyper-parameter search for optimal CFG schedules and dynamic CFG, we show via ablations that there is no globally optimal CFG schedule even for a specific model. In Table 2, we compare heuristic schedules from the literature (limited guidance interval, annealing) and even our automatically derived average schedule across prompts and show that only dynamic CFG is able to improve on both alignment and visual quality at the same time highlighting the importance to optimize CFG per prompt. Moreover, we empirically demonstrate in Figure 3 how different prompts require different schedules with lower or higher guidance depending on the generation needs (e.g., creative prompts vs. text rendering and compositionality).
>
> We overall challenge prior claims that treat CFG as static and draw specific conclusions, such as that low guidance scales are necessary in the initial denoising steps for good quality generations, by showing that such claims do not transfer to other model families and they cannot even be applied for all prompts within a specific model family. **Our framework is model-agnostic since it can successfully find optimal CFG schedules that on average agree with the cumbersome manual hyper-parameter tuning per model family without adding any model-specific bias to the search algorithm.**

---

> > ### Author Response · Authors · 2025-11-20
> > **Questions**
> >
> > - _Lines 216–220 could be misread as computing all CFG outputs in parallel and assembling them afterward. Could this part be explained better?_
> >
> > See response to Weakness point 1.
> >
> > - _Section 5.1 is named Evaluation of latent evaluators, but in practice only evaluates the performances of the latent CLIP evaluator and not of the others. Would it be possible to provide a justification of this choice?_
> >
> > We also provide results for the discriminator in Appendix A.5, but we omitted these results from the main paper due to space limitations. We mention in Line 316 that we observe a similar behavior for the visual quality evaluator. Overall, we ablated the evaluators on LDM small and large and used both evaluators that were also used for our main results in Table 2.
> >
> > - Equation 6 refers to W which, if I am not wrong, is never defined before. May this be S?
> >
> > Yes, this is correct. This was a typo and we have now updated the paper.
> >
> > - In Equation 7 the alpha coefficients could potentially assume negative values. Should there be an absolute value around the difference between et and et+1?
> >
> > We do not use the absolute value of the difference of the evaluator scores. We observe that the difference is positive for most cases, since typically for lower noise levels the average evaluator scores increase across timesteps even for low alignment. However, we ablated replacing the difference with the absolute value and results were inferior, which indicates that getting negative weighting on evaluator scores might penalize the evaluator’s predictions.

---

### Official Review · Reviewer_3AEj · 2025-10-31

**Soundness:** 3
**Presentation:** 2
**Contribution:** 2
**Rating:** 6
**Confidence:** 2

**Summary:**

* The paper introduces a framework for dynamic CFG in T2I diffusion models.
Unlike conventional static CFG scales or fixed heuristic schedules, the proposed method adaptively adjusts the CFG scale at each sampling step using online feedback from multiple evaluators.
* These evaluators include a CLIP-based alignment model, a discriminator for visual fidelity, a human preference reward model, and specialized evaluators for text rendering and numerical reasoning.
* The framework employs a greedy search to select the optimal CFG scale per timestep, generating a prompt-dependent schedule.
* Experiments on both SD models (LDM) and the Imagen 3 show consistent improvements across alignment, visual quality, text rendering, and counting tasks.

**Strengths:**

* The idea of making CFG dynamic and feedback-driven is interesting.
It challenges the assumption that CFG scales should remain static during generation.
* Operating in the latent space keeps the method lightweight, making it feasible for real-world use.
* The framework transfers successfully from diffusion models to large-scale proprietary ones like Imagen 3, showing robustness and adaptability.
* The authors assess their method across diverse benchmarks combining quantitative metrics and human preference studies.
* Visualizations of dynamic schedules clearly show how guidance strength varies with prompt complexity, enhancing transparency and interpretability of the approach.

**Weaknesses:**

* I believe per-step greedy search does not guarantee global optimality.
The method assumes independence between timesteps, which may overlook temporal correlations in the diffusion trajectory.
* The comparisons may be a bit limited.
Including stronger recent baselines, like RL-based or multi-step optimization methods (FK steering or diffusion RL), would make the results more convincing.
* In my understanding, each evaluator must be re-trained for the specific diffusion model, which weakens the claim of being fully model-agnostic.

**Questions:**

* How does the method ensure temporal consistency of guidance across timesteps given the greedy selection strategy?
* Could a differentiable or policy-gradient optimization approach yield smoother and potentially superior CFG schedules?

---

> ### Author Response · Authors · 2025-11-20
> **Weakness Point 1**
>
> _I believe per-step greedy search does not guarantee global optimality. The method assumes independence between timesteps, which may overlook temporal correlations in the diffusion trajectory._
>
> Indeed, a greedy selection of the best CFG scale per timestep does not guarantee a global maximum of the quality of generated samples. However, based on the analysis of our evaluators in Section 5.1 as we also explained above, we find that the evaluators are good predictors of final sample quality (i.e., they can predict alignment from as early as 25% during denoising and prediction after 75% of denoising remains the same).
>
> Our greedy search is similar to LLM decoding strategies. The most efficient (and simplest) method in LLM decoding is greedy search, which similarly does not guarantee that the final generated sentence is the most probable but it can be used as an approximation.
>
> Alternatively, **we tried an approach similar to beam search where at each denoising step we keep the top-k candidates instead of considering the maximum score only**. For each candidate i at timestep t, we sample x^i_{t-1} considering a set of different CFG values. We next select the top-k CFG values per x^i_{t-1}. Finally, we prune the candidates across x^1_{t-1} , x^2_{t-1},..., x^k_{t-1} in order to only keep k candidates for the following denoising step.
>
> The biggest downside of this approach is the computational cost, as we need to sample k times per timestep from the diffusion model before exploring different CFG values for each candidate (in contrast to greedy search that does not increase the NFEs). We compared this approach against greedy search during early experimentation on Imagen 3 on the Gecko prompt set. Specifically, **we compared greedy search against beam search, when k=2 which doubles the NFEs required for sampling**. For our ablation, we used the latent alignment evaluator (CLIP) and we automatically measured final text-image alignment of the samples via the Gecko metric. **We find that greedy search achieves 73.1% on the Gecko prompt set in contrast to 73.7% for beam search**. We concluded that the **improvement when using beam search is negligible with respect to the increased computational requirements**. We will add this ablation/discussion in the Appendix.

---

> > ### Author Response · Authors · 2025-11-20
> > **Weakness Point 2**
> >
> > _The comparisons may be a bit limited. Including stronger recent baselines, like RL-based or multi-step optimization methods (FK steering or diffusion RL), would make the results more convincing._
> >
> > As we mention in Section 2, we consider our work orthogonal to methods like FK steering, since they start from multiple initial seeds and significantly increase NFEs and hence the inference-time computational cost (especially for SoTA denoisers like Imagen). In contrast, **our method aims at improving generation from a single seed without increasing NFEs and with a minimal 1% computational overhead during sampling**.
> >
> > Although FK steering and our approach are not comparable due to the different computational requirements, we ablated their relative performance on Imagen 3 to understand the tradeoff between increasing compute (when considering multiple seeds) and  sample quality. We implemented the FK steering algorithm for sampling from Imagen 3 when considering k=4 particles.
> >
> > We test FK steering and our dynamic CFG on the Gecko prompt set, which is a general-purpose prompt set containing diverse prompts for different generation capabilities, using the latent reward evaluator. We next ran human evaluation for comparing FK steering and dynamic CFG and measured the win rate of FK steering against our method (similar to the results reported in Table 3). We find that the **win rate for FK steering is 50.7%** with a **confidence interval of [48.9%, 52.5%]**, which indicates that the **two methods' performance difference is not statistically significant**– interestingly **despite more compute, there is no significant advantage for FK steering**.
> >
> >
> > | Comparison         |                            Win Rate ↑ (Confidence Interval) |
> > | ---------------------------------- |:------------------------------------------:|
> > |Dynamic CFG vs FK steering      |     50.7% ([48.9%, 52.5%])|
> >
> >
> >
> > This result further strengthens the validity of our approach since we show that dynamic optimization of CFG for a fixed initial seed with only 1% computational overhead and no increase in NFEs is comparable to inference-time scaling methods such as FK steering that search over multiple initial seeds and significantly increase test-time computation. We will add a section with the results and discussion in the main paper upon acceptance.
> >
> > Regarding comparisons with diffusion RL approaches:
> >
> > - We consider that our inference-time approach is orthogonal to post-training methods and targets per-prompt optimization without changing model weights. **Our results on Imagen 3 are on top of post-training showing that our approach can improve sample quality in addition to post-training**. During early experimentation on intermediate model versions, we found that we consistently improve sampling under different training and fine-tuning regimes.
> >
> > - **The resources required for adapting the evaluators on the latent-space and/or specialized datasets are orders of magnitude lower than fine-tuning large SoTA diffusion models**. We provide the exact training requirements for the evaluators in Appendix A.1. Note that , we do not train the evaluators from scratch, but we adapt a general-purpose CLIP model to the diffusion latent space by further training the model on 64 TPUv5e (16GB) chips for 1.5 days (i.e., latent alignment evaluator of Equation 1). All remaining specialized evaluators are fine-tuned starting from the latent CLIP model for an additional 10k steps on 8 TPUv5e chips for 8 hours. This is orders of magnitude cheaper than fine-tuning large diffusion models in terms of computational costs.
> >
> > - Our approach involves short fine-tuning of small-scale latent evaluators which is **straight-forward and stable**. In contrast, RL-based diffusion approaches are much more challenging and can present instabilities during training.

---

> ### Author Response · Authors · 2025-11-20
> **Weakness Point 3**
>
> _In my understanding, each evaluator must be re-trained for the specific diffusion model, which weakens the claim of being fully model-agnostic._
>
> The latent evaluators depend on the diffusion latent space, so indeed they need to be retrained for different diffusion autoencoders. From this aspect, **the evaluators are not model-agnostic**. However, we show in the paper that **our framework is model-agnostic**, as our results transfer from a smaller open source latent diffusion model to SoTA image generation after post-training. What is interesting is that our framework can automatically find different patterns of CFG schedules for different model families: as shown in Figure 2, for LDM our framework proposes dynamic CFG schedules that on average agree with empirical observations on best CFG values for Stable Diffusion models. However, using the same exact framework and algorithm while adapting the evaluators, we see that the proposed dynamic CFG schedules for Imagen 3 are on average very different from our LDM results and empirical observations from the literature. Hence, we conclude that prior work that analyses the CFG schedules and claims agnostic patterns in generation cannot generalize to different model families. In contrast, our framework can retrieve optimal CFG schedules for different model families (given of course the fine-tuned evaluators that match the latent space).
>
> We overall challenge prior claims that treat CFG as static and draw specific conclusions, such as that low guidance scales are necessary in the initial denoising steps for good quality generations, by showing that such claims do not transfer to other model families and they cannot even be applied for all prompts within a specific model family. **Our framework is model-agnostic since it can successfully find optimal CFG schedules that on average agree with the cumbersome manual hyper-parameter tuning per model family without adding any model-specific bias to the search algorithm.**

---

> > ### Author Response · Authors · 2025-11-20
> > **Questions**
> >
> > - _How does the method ensure temporal consistency of guidance across timesteps given the greedy selection strategy?_
> >
> > See response to Weakness Point 1.
> >
> > - _Could a differentiable or policy-gradient optimization approach yield smoother and potentially superior CFG schedules?_
> >
> > Instead of using greedy search, we have also **experimented with a differentiable optimization variant for our dynamic CFG approach**.
> >
> > Specifically, for each timestep t, instead of searching over a set S of candidate CFG values, we compute the gradient of the evaluator e_t with respect to the CFG value s considering the initial x_t derived by using a default CFG value s_{init}: gradient_s(e_t(x_t^s_{init}) and then update sample the optimal CFG value s according to the gradient: s = s_init + lambda*gradient_s(e_t(x_t^s_{init})) and re-compute xt with respect to the updated s. We have experimented with correcting the CFG value s per timestep independently by always considering the same default s_{init} or through time, by also updating the default CFG value after each timestep.
> >
> > **We compared the two methods (gradient vs search) during initial experimentation on Imagen 3**. We measured text-image alignment automatically via the Gecko score on the Gecko prompt set. **We found that search performs slightly better than gradient (80.8% vs  78.9%, respectively). But more importantly, tuning the gradient approach (in terms of s_{init} and lambda)  was much more strenuous and small changes in these hyper-parameters would cause significant drops in accuracy.** For this reason, we selected search as the most stable and robust approach, which does not depend on cumbersome hyper-parameter tuning, for dynamic CFG.
> >
> > We did not experiment with policy-gradient optimization, as this direction is generally more challenging and unstable (e.g., Section C in [1]).
> >
> > [1] Xue, Zeyue, et al. "DanceGRPO: Unleashing GRPO on Visual Generation." arXiv preprint arXiv:2505.07818 (2025).

---

### Official Review · Reviewer_ik4p · 2025-10-31

**Soundness:** 3
**Presentation:** 4
**Contribution:** 2
**Rating:** 6
**Confidence:** 4

**Summary:**

This paper proposes an approach, Dynamic CFG, for dynamically choosing the classifier-guidance scale at each individual timestep using a series of online evaluators. By choosing the optimal cfg-scale with respect to the alignment/visual quality/human preference score, Dynamic CFG improved overall generation quality with adaptive evaluators’  weight method.

**Strengths:**

**Clear motivation and well-written:** The proposed method is clear motivated and the writing of this paper is great. Most of terms have been clearly explained. It is conformable to this paper.

**Lightweight evaluator design:** The latent evaluator is lightweight and can be seamlessly integrated into existing diffusion models without significantly increasing computational cost.

**Strong effectiveness:** The method demonstrates consistent improvements across multiple benchmarks, and it could enhance text alignment, visual quality, and human preference win rates on strong baselines like LDM and Imagen 3.

**Weaknesses:**

**High training cost for evaluators:** The method requires training multiple specialized latent evaluators with dedicated training datasets. This increases the overall implementation cost and limits the practicality of applying the approach to new tasks (e.g., when user wants to enhance the layout alignment).

**Dependence on evaluator quality:** The performance of the proposed method heavily relies on the accuracy and generalization ability of the evaluators. One poorly trained evaluator may lead to unstable or suboptimal guidance schedules. The authors could add more discussion for this question to increase the transparency of the proposed method.

**Questions:**

1. While the online evaluator is lightweight and the inference cost is low, and the current choice of five CFG-scale candidates proves effective, it remains unclear how much additional performance gain could be achieved by introducing more candidate scales.

2. Transferability: For the latent evaluator, can it be transferred to other models trained in the same latent space?

---

> ### Author Response · Authors · 2025-11-20
> **Weakness Point 1**
>
> _High training cost for evaluators: The method requires training multiple specialized latent evaluators with dedicated training datasets. This increases the overall implementation cost and limits the practicality of applying the approach to new tasks (e.g., when user wants to enhance the layout alignment)._
>
> **All evaluators are based on the latent alignment evaluator (CLIP)** which uses a standard CLIP architecture without implementation differences.
>
> - One important step of our approach is to **adapt a general-purpose CLIP model** to the diffusion latent space. This requires fine-tuning of the general off-the-shelf CLIP model, for which we report exact resource requirements in **Appendix A.1 (64 TPUv5e chips for 1.5 days)**. However, for this adaptation, we use the **same general image-text data from the WebLI dataset** that was originally used for training the CLIP model. Interestingly, **this approach seems to generalize well from the WebLI data to the prompts** given to gLDM and Imagen 3 for sampling, which are significantly different. Moreover, using this more general-purpose CLIP evaluator for alignment seems to **even generalize well for different generation capabilities** as it can significantly improve performance on the text rendering prompt set without specialized training as reported in Table 3 (52.3% win rate against the baseline).
>
> - We do look at the **applicability of our framework for new tasks (e.g. Text Rendering and Numerical Reasoning)** that, similarly to spatial reasoning, require complex reasoning over the generated image and are able to improve results by a significant margin than the baseline model. For these evaluators, we use the above latent CLIP model as our base model and **we only fine-tune it for a few more steps (10k) on 8 TPUv5e chips for 8 hours on the smaller dedicated datasets (details in Appendix A.1)**. As dedicated datasets, we use human preference data when available (i.e., reward evaluator), but **we also automatically curate task-specific data** for text rendering (via OCR scores) and for numerical reasoning (via synthetic captions by Gemini 2.5 Pro) showing that different data sources can work effectively for our framework. We show that for specific use cases, where we might care more for specific generation aspects and/or have curated or human-collected data, (1) our framework can generalize to these specific generation aspects, and (2) we can improve sampling with respect to the specific skill further than general-purpose evaluators.

---

> > ### Author Response · Authors · 2025-11-20
> > **Weakness Point 2**
> >
> > _Dependence on evaluator quality: The performance of the proposed method heavily relies on the accuracy and generalization ability of the evaluators. One poorly trained evaluator may lead to unstable or suboptimal guidance schedules. The authors could add more discussion for this question to increase the transparency of the proposed method._
> >
> >
> > Indeed, the quality of the evaluators is important for improving sampling with our method. However, we show that **even a general-purpose CLIP model adapted to the diffusion latent space without being further trained on more curated datasets (i.e., the latent alignment evaluator) can offer significant improvements** in the dynamic CFG approach (see Table 3). Interestingly, the data distribution of the WebLI dataset that the CLIP model is trained on, is very noisy and significantly different from the prompts we are using for inference (e.g., length, level of description of a scene). This general latent alignment evaluator not only improves performance on a general-purpose prompt set (53.2% win rate on GenAI-Bench) but even generalizes to specific prompt sets: it significantly improves text rendering in generated images as measured in the MARIO-eval prompt set (52.3% win rate) without any specialized training. These all show that **while our approach depends on a high quality evaluator, training a general-purpose evaluator that can significantly improve sampling is straightforward**.
> > Nevertheless, we do observe further improvements in the evaluator’s performance by improving the training data. For example, further training on text rendering specific data can improve performance on MARIO-eval even further (up to 55.5%). We will add this discussion in the paper.

---

> > > ### Author Response · Authors · 2025-11-20
> > > **Questions**
> > >
> > > - _While the online evaluator is lightweight and the inference cost is low, and the current choice of five CFG-scale candidates proves effective, it remains unclear how much additional performance gain could be achieved by introducing more candidate scales._
> > >
> > > 1. We use five CFG scale candidates for LDM and we find that increasing the number of candidates does not improve results, possibly due to the lower model quality. However, we use 24 candidates for dynamic CFG on Imagen 3, which seems to be benefiting more from a larger set of candidates. We will update the paper to specify this.
> > >
> > > 2. During early experimentation, we considered different sets of CFG scale candidates to select from for Imagen 3. We ablated different sets on the Gecko prompt set using the latent alignment evaluator (CLIP). In all cases we measured alignment of the final samples automatically via the Gecko metric. We provide the results of the ablation below. Overall, we see that there is a small improvement in performance by including more candidates, but this performance gain is capped at 24 candidates.
> > >
> > > | # CFG scale candidates | Gecko score (alignment) ↑ |
> > > | :----------------------------:| :----------------------------:|
> > > | 16 | 72.5 |
> > > | 19 | 72.8 |
> > > | 21 | 73.1 |
> > > | 24 | 74.1 |
> > > | 27 | 74.0 |
> > >
> > > We will update the Appendix to include this ablation.
> > >
> > > - _Transferability: For the latent evaluator, can it be transferred to other models trained in the same latent space?_
> > >
> > > Yes, the latent evaluator only depends on the encoder used for generating the latents. For example, LDM small and large both use the same latent space and therefore we only train the evaluators once and use them for both model versions.

---

### Official Review · Reviewer_67qG · 2025-10-31

**Soundness:** 1
**Presentation:** 2
**Contribution:** 1
**Rating:** 2
**Confidence:** 4

**Summary:**

This paper introduces a dynamic Classifier-Free Guidance (CFG) scheduling framework for text-to-image diffusion models, replacing the traditional static, one-size-fits-all guidance scale with adaptive, prompt-specific schedules. The method uses latent-space evaluators (e.g., CLIP, discriminator, and human preference reward models) to provide real-time feedback on alignment, fidelity, and reasoning quality during the diffusion process. A greedy search dynamically selects the optimal CFG scale at each timestep, yielding schedules that adapt to both the prompt and diffusion stage. Experiments on StableDiffusion-like models and Imagen 3 show substantial gains in text alignment, visual quality, and human preference, proving that optimal CFG is inherently dynamic and prompt-dependent.

**Strengths:**

The authors identify a interesting research question: how to improve generated images across multiple aspects such as alignment, fidelity, et, by dynamically adapting the Classifier-Free Guidance (CFG) scale instead of relying on a static, fixed value.

**Weaknesses:**

1. The approach heavily depends on CLIP image encoder fine-tuning under noisy latent conditions, which may distort the original embedding space; however, the paper provides neither validation nor ablation studies to justify or quantify this effect.

2. The experimental comparison is limited to conventional baselines. For example, [1] introduces a test-time optimization strategy designed for multi-reward settings. The authors may need to compare their method with this. In additions, considering that the method employs multiple specialized evaluators (e.g., human preference, counting) and substantial fine-tuning, it can also be compared against multi-objective RL-based diffusion approaches, which incur similar computational costs.

3. The assumption that a greedy-selection-based dynamic CFG universally enhances performance across different aspects lacks both theoretical grounding and intuitive justification; additional analysis or formal reasoning is necessary.

4. The overall writing quality could be improved for clarity, coherence, and readability.

[1] Test-time Alignment of Diffusion Models without Reward Over-optimization, Kim et al., ICLR 2025

**Questions:**

It is unclear what $W$ represents in Equation (6); the authors should explicitly define it.

The relationship between the guidance scale $s$ and the evaluator score $e_t$  needs clarification. Is $e_t$ defined as an aggregate across multiple evaluators?
If $e_t$ is indeed defined as the average over different evaluator scores, this should be explicitly stated, along with the precise formulation.

The notation $E$ in Equation (7) is also ambiguous. The authors should clarify this as well.

---

> ### Author Response · Authors · 2025-11-20
> **Weakness Point 1**
>
> _The approach heavily depends on CLIP image encoder fine-tuning under noisy latent conditions, which may distort the original embedding space; however, the paper provides neither validation nor ablation studies to justify or quantify this effect._
>
> We have provided both validation and ablations in **Section 5.1** of the main paper and Appendix.  While using an encoder over noisy latent conditions could lead to information loss in contrast to using the final clean images or the corresponding pixel space images obtained during sampling, we show that we can achieve a good trade off between performance and efficiency.
> In particular, we compare the relative information loss:
>
> (1) at different noise levels starting from 75% noised, and
>
> (2) comparing latent-space and pixel-space evaluators at different degrees of noise. In order to quantify this effect, we use the different latent- or pixel-space evaluators for filtering bad samples at different stages during denoising and we measure text-image alignment via the Gecko metric.
>
> **We discuss this in Section 5.1 where we evaluate our evaluators and present results in Table 1**. What we find overall is:
>
> 1. Operating in the latent space leads to minimal information loss in contrast to pixel-space: a **latent-space model improves** filtering performance **by 8.6%** for LDM large in contrast to a **pixel-space** equivalent that improves performance **by 12.8%**.
>
> 2. We  can **successfully measure text-image alignment as early as 25%** (when the images are highly noisy) during denoising (where filtering performance increases over the baseline) and we notice no significant improvements after 75% of denoising.
>
> 3. The behavior and patterns are consistent for different model sizes (LDM small and large) and interestingly for different noise levels when comparing latent and pixel space evaluators.
>
> Given these results, it is clear that leveraging CLIP fine-tuned on noisy latents is effective and as predictive as using intermediary images at a fraction of the cost (~**4 times less compute** when operating in the latent instead of the pixel space according to the FLOPs reported for LDM in **Table 5 and discussed in Section A.3 of the Appendix**).

---

> > ### Author Response · Authors · 2025-11-20
> > **Weakness Point 2**
> >
> > _The experimental comparison is limited to conventional baselines. For example, [1] introduces a test-time optimization strategy designed for multi-reward settings. The authors may need to compare their method with this. In additions, considering that the method employs multiple specialized evaluators (e.g., human preference, counting) and substantial fine-tuning, it can also be compared against multi-objective RL-based diffusion approaches, which incur similar computational costs._
> >
> > We thank the reviewer for providing the reference, it is indeed relevant to our work and we have updated the manuscript to cite the paper appropriately. The method suggested by the reviewer is a test-time optimization method similar to Feynman Kac (FK) steering ([1], already cited in our work). As we mention in Section 2, we consider our work orthogonal to these methods, since DAS and FK steering start from multiple initial seeds and significantly increase NFEs and hence the inference-time computational cost (especially for SoTA denoisers like Imagen). In contrast, our method aims at improving generation from a single seed without increasing NFEs and with a minimal 1% computational overhead during sampling.
> >
> > Although DAS/FK steering and our approach are not comparable due to the different computational requirements, we ablated their relative performance on Imagen 3 to understand the tradeoff between increasing compute (when considering multiple seeds) and  sample quality. We implemented the FK steering algorithm for sampling from Imagen 3 when considering k=4 particles.
> >
> > We test FK steering and our dynamic CFG on the Gecko prompt set, which is a general-purpose prompt set containing diverse prompts for different generation capabilities, using the latent reward evaluator. We next ran human evaluation for comparing FK steering and dynamic CFG and measured the win rate of FK steering against our method (similar to the results reported in Table 3). We find that the **win rate for FK steering is 50.7%** with a **confidence interval of [48.9%, 52.5%]**, which indicates that the **two methods' performance difference is not statistically significant**– interestingly **despite more compute, there is no significant advantage for FK steering**.
> >
> >
> > | Comparison         |                            Win Rate ↑ (Confidence Interval) |
> > | ---------------------------------- |:------------------------------------------:|
> > |Dynamic CFG vs FK steering      |     50.7% ([48.9%, 52.5%])|
> >
> >
> >
> > This result further strengthens the validity of our approach since we show that dynamic optimization of CFG for a fixed initial seed with only 1% computational overhead and no increase in NFEs is comparable to inference-time scaling methods such as FK steering that search over multiple initial seeds and significantly increase test-time computation. We will add a section with the results and discussion in the main paper upon acceptance.
> >
> > Regarding comparisons with diffusion RL approaches:
> >
> > - We consider that our inference-time approach is orthogonal to post-training methods and targets per-prompt optimization without changing model weights. **Our results on Imagen 3 are on top of post-training showing that our approach can improve sample quality in addition to post-training**. During early experimentation on intermediate model versions, we found that we consistently improve sampling under different training and fine-tuning regimes.
> >
> > - **The resources required for adapting the evaluators on the latent-space and/or specialized datasets are orders of magnitude lower than fine-tuning large SoTA diffusion models**. We provide the exact training requirements for the evaluators in Appendix A.1. Note that , we do not train the evaluators from scratch, but we adapt a general-purpose CLIP model to the diffusion latent space by further training the model on 64 TPUv5e (16GB) chips for 1.5 days (i.e., latent alignment evaluator of Equation 1). All remaining specialized evaluators are fine-tuned starting from the latent CLIP model for an additional 10k steps on 8 TPUv5e chips for 8 hours. This is orders of magnitude cheaper than fine-tuning large diffusion models in terms of computational costs.
> >
> > - Our approach involves short fine-tuning of small-scale latent evaluators which is **straight-forward and stable**. In contrast, RL-based diffusion approaches are much more challenging and can present instabilities during training.
> >
> > [1] Raghav Singhal, Zachary Horvitz, Ryan Teehan, Mengye Ren, Zhou Yu, Kathleen McKeown, and Rajesh Ranganath. A general framework for inference-time scaling and steering of diffusion models. In Forty-second International Conference on Machine Learning, 2025.

---

> ### Author Response · Authors · 2025-11-20
> **Weakness Point 3**
>
> _The assumption that a greedy-selection-based dynamic CFG universally enhances performance across different aspects lacks both theoretical grounding and intuitive justification; additional analysis or formal reasoning is necessary._
>
> Indeed, a greedy selection of the best CFG scale per timestep does not guarantee a global maximum of the quality of generated samples. However, based on the analysis of our evaluators in Section 5.1 as we also explained above, we find that the evaluators are good predictors of final sample quality (i.e., they can predict alignment from as early as 25% during denoising and prediction after 75% of denoising remains the same).
>
> Our greedy search is similar to LLM decoding strategies. The most efficient (and simplest) method in LLM decoding is greedy search, which similarly does not guarantee that the final generated sentence is the most probable but it can be used as an approximation.
>
> Alternatively, **we tried an approach similar to beam search where at each denoising step we keep the top-k candidates instead of considering the maximum score only**.
> For each candidate i at timestep t, we sample x^i_{t-1} considering a set of different CFG values. We next select the top-k CFG values per x^i_{t-1}. Finally, we prune the candidates across x^1_{t-1} , x^2_{t-1},..., x^k_{t-1} in order to only keep k candidates for the following denoising step.
>
> The biggest downside of this approach is the computational cost, as we need to sample k times per timestep from the diffusion model before exploring different CFG values for each candidate (in contrast to greedy search that does not increase the NFEs). We compared this approach against greedy search during early experimentation on Imagen 3 on the Gecko prompt set. Specifically, **we compared greedy search against beam search, when k=2 which doubles the NFEs required for sampling**. For our ablation, we used the latent alignment evaluator (CLIP) and we automatically measured final text-image alignment of the samples via the Gecko metric. **We find that greedy search achieves 73.1% on the Gecko prompt set in contrast to 73.7% for beam search**. We concluded that **the improvement when using beam search is negligible with respect to the increased computational requirements**. We will add this ablation/discussion in the Appendix.

---

> > ### Author Response · Authors · 2025-11-20
> > **Questions**
> >
> > - _It is unclear what W represents in Equation (6); the authors should explicitly define it.
> > The relationship between the guidance scale s and the evaluator score et needs clarification. Is et defined as an aggregate across multiple evaluators? If et is indeed defined as the average over different evaluator scores, this should be explicitly stated, along with the precise formulation._
> >
> > We have updated the manuscript accordingly.
> > Equation 6 has a typo and we thank the reviewer for spotting this. It is the set S of CFG scales and not W.
> > We also updated Equation (6) to reflect the relationship between s and et more clearly: we compute the evaluator’s score et for each variant of x^s_t for each different scale s.
> > et can be the score of a single evaluator or can be the aggregate when multiple evaluators are used for inference. In the latter case, we either use a simple linear weighted average of the evaluators score as presented in Table 2 or we propose to define an adaptive aggregate depending on the timestep and evaluators’ scores as defined in Equation 7.
> >
> > - _The notation E in Equation (7) is also ambiguous. The authors should clarify this as well._
> >
> > E is the set of evaluators that we use. We have updated the manuscript.

---

### Comment · Area_Chair_jyeS · 2025-11-27
**Please check the rebuttal**

Dear Reviewers,

The authors have posted their rebuttal. Could you please take a moment to review their responses and check whether your concerns have been adequately addressed if you have done it yet? If possible, kindly initiate the discussion at your earliest convenience.

Your timely assistance is essential for keeping the review process on track. Thank you very much for your support and contribution.

Best regards,
Your AC

---

### Meta-Review · Area_Chair_EbRD · 2026-01-07

**Summary:**

The paper proposes a method to dynamically adjust the scale of classifier-free guidance (CFG) in latent diffusion models for text-to-image generation. The key idea is to learn lightweight evaluators that operate in the latent space and to greedily select the best guidance scale from a candidate set at each reverse diffusion step. The authors present extensive experiments to demonstrate the proposed method.

The initial reviews were polarized. Reviewers found the idea interesting, yet raised several concerns, though some of these comments were not well grounded. The common issues across reviewers included: (1) if per-step greedy selection can lead to a globally optimal solution; (2) if the dependence between the evaluators and the diffusion model undermines the model-agnostic claim; and (3) the lack of comparisons to more recent methods, e.g., FK steering.

The authors' rebuttal addressed these concerns and provided additional technical clarification.

After reviewing the paper, reviews, and rebuttal, the AC agrees that the main idea and findings are interesting and worth communicating to the community. The AC therefore recommends acceptance. The authors are encouraged to further incorporate reviewer feedback and refine the paper.

**Reviewer Concerns:**

The common concerns were well addressed in the rebuttal.
* The authors justified the design of per-step greedy selection based on existing empirical analysis. They also added a smaller-scale experiment and compare to beam search, where greedy selection remain competitive.
* The authors clarified that while the evaluators themselves are not model-agnostic, the overall framework can be applied across different diffusion models, though further clarification in the paper would be beneficial.
* The authors added a new experimental comparison with FK steering and demonstrated comparable performance.

The authors also provided satisfactory responses to the remaining valid concerns. The AC did not find major outstanding issues.

**Reviewer Scores:**

Most reviewers are likely to maintain or raise their scores if the rebuttal is taken into consideration.

---

### Decision · Program_Chairs · 2026-01-26

Accept (Poster)